# High Intakes of Bioavailable Phosphate May Promote Systemic Oxidative Stress and Vascular Calcification by Boosting Mitochondrial Membrane Potential—Is Good Magnesium Status an Antidote?

**DOI:** 10.3390/cells10071744

**Published:** 2021-07-09

**Authors:** Mark F. McCarty, Aaron Lerner, James J. DiNicolantonio, Simon B. Iloki-Assanga

**Affiliations:** 1Catalytic Longevity Foundation, 811 B Nahant Ct., San Diego, CA 92109, USA; markfmccarty@gmail.com; 2Chaim Sheba Medical Center, The Zabludowicz Research Center for Autoimmune Diseases, Tel Hashomer 5262000, Israel; 3Mid America Heart Institute, St. Luke’s Hospital, Kansas City, MO 64111, USA; jjdinicol@gmail.com; 4Department of Research and Postgraduate in Food Science, Universidad de Sonora, Hermosillo 83000, Mexico; ilokiasb@yahoo.com

**Keywords:** phosphate, calcium, mitochondria, oxidative stress, fibroblast growth factor 23, magnesium

## Abstract

Chronic kidney disease is characterized by markedly increased risk for cardiovascular mortality, vascular calcification, and ventricular hypertrophy, and is associated with increased systemic oxidative stress. Hyperphosphatemia, reflecting diminished glomerular phosphate (Pi) clearance, coupled with a compensatory increase in fibroblast growth factor 23 (FGF23) secretion are thought to be key mediators of this risk. Elevated serum and dietary Pi and elevated plasma FGF23 are associated with increased cardiovascular and total mortality in people with normal baseline renal function. FGF23 may mediate some of this risk by promoting cardiac hypertrophy via activation of fibroblast growth factor receptor 4 on cardiomyocytes. Elevated serum Pi can also cause a profound increase in systemic oxidative stress, and this may reflect the ability of Pi to act directly on mitochondria to boost membrane potential and thereby increase respiratory chain superoxide production. Moreover, elevated FGF23 likewise induces oxidative stress in vascular endothelium via activation of NADPH oxidase complexes. In vitro exposure of vascular smooth muscle cells to elevated Pi provokes an osteoblastic phenotypic transition that is mediated by increased mitochondrial oxidant production; this is offset dose-dependently by increased exposure to magnesium (Mg). In vivo, dietary Mg is protective in rodent models of vascular calcification. It is proposed that increased intracellular Mg opposes Pi’s ability to increase mitochondrial membrane potential; this model could explain its utility for prevention of vascular calcification and predicts that Mg may have a more global protective impact with regard to the direct pathogenic effects of hyperphosphatemia.

## 1. Introduction

Mitochondrial diseases are a group of conditions in which mitochondrial dysfunction leads to cellular failure to generate sufficient ATP to maintain cellular homeostasis, and/or mitochondrial oxidant generation adversely affects the function or survival of cells [1]. Mitochondrial function is compromised in rare monogenic and very common polygenic diseases, including metabolic, cardiovascular, neurodegenerative, neuroinflammatory, and neuromuscular disorders, as well as cancer [2,3,4]. A common dominator in these chronic conditions is oxidative stress and vascular dysfunction, as seen in cardiovascular [5] and renal diseases [6].

While dietary phosphate (Pi) is self-evidently essential for health, chronically elevated Pi levels associated with impaired glomerular function and episodic elevations reflecting diets rich in bioavailable Pi—from animal products and Pi food additives—are associated with increased cardiovascular risk characterized by vascular calcification [7,8]. Conversely, increased dietary intakes or serum levels of magnesium (Mg) have been correlated with reduced risk of cardiovascular events and vascular calcification [9,10,11]. The thesis presented here is that improved Mg status can function as a detoxicant for Pi excess by opposing the tendency of elevated Pi to promote excess oxidant production by mitochondria. Within vascular smooth muscle, this oxidative stress promotes a phenotypic transition giving rise to vascular calcification; in vascular endothelium, it can compromise the cardiovascular protection conferred by a healthy endothelium.

## 2. Cardiovascular Toxicity of Elevated Phosphate and Fibroblast Growth Factor 23

Chronic uremia is associated with greatly increased risk for cardiovascular events and mortality, and most notably with high risk for vascular calcification. Hyperphosphatemia, reflecting subnormal renal phosphate clearance, is a primary mediator of the cardiovascular risk associated with uremia. While the cardiotoxic effect of hyperphosphatemia has been best characterized in the context of chronic renal failure, prospective studies in healthy cohorts or in patients with coronary ischemic heart disease have found that cardiovascular mortality risk correlates positively with serum Pi; additionally, in the NHANES III cohort, calorie-corrected Pi dietary intake correlated directly with cardiovascular and total mortality [12,13,14,15]. One recent study has, however, found that individuals with the lowest plasma Pi are also at increased risk in this regard, a finding not reported by previous studies [16]. Ingestion of a high-Pi meal has been shown to acutely impair endothelium-dependent vasodilation in healthy human subjects [17].

Elevations of plasma phosphate evoke increased secretion of fibroblast growth factor 23 (FGF23) from osteoblasts, which acts on the kidneys to suppress reabsorption of Pi by proximal kidney tubules, while also inhibiting renal conversion of calcidiol (25-hydroxyvitamin D) to the active hormone calcitriol; the latter effect serves to decrease intestinal Pi absorption [18,19]. Hence, FGF23 acts homeostatically to lower plasma Pi levels when these levels rise. Unfortunately, the increased secretion of FGF23 associated with uremia also contributes to the cardiovascular risk associated with this syndrome. Elevated FGF23 predicts increased cardiovascular and total mortality in patients with chronic kidney disease [20,21]. This phenomenon has now been reported in healthy populations, independent of glomerular filtration rate; in one recent prospective study, individuals in the top quartile of FGF23, in comparison to those in the bottom quartile, were found to have double the risk for cardiovascular mortality and about a 60% greater risk for non-cardiovascular mortality, after adjustment for relevant covariates (including serum phosphate) [22,23]. Evidently, some of this risk might be mediated by increased dietary Pi intake (fasting serum Pi is not directly proportionate to dietary Pi absorption, so statistical correction for fasting Pi may not eliminate the full impact of dietary Pi), and the lower risk associated with low-normal FGF23 might also reflect the protection inherently afforded by plant-based diets, which tend to be lower in bioavailable Pi. Although plant-based diets are not low in Pi per se, much of their Pi content usually comes from phytates which cannot be degraded by human digestive enzymes, whereas the Pi found in animal products tends to have high bioavailability [24,25,26].

Nonetheless, there is good reason to suspect that FGF23 itself mediates much of the increased risk associated with elevated plasma levels of this hormone. FGF23 has a direct impact on the heart that promotes ventricular hypertrophy acting via FGF receptor 4 (FGFR4) in cardiomyocytes [27,28,29]. Whereas FGF23 can only activate other forms of the FGF receptor in conjunction with the co-receptor α-klotho (not expressed in cardiomyocytes), FGF23 can activate FGFR4 in the absence of this co-receptor, leading to autophosphorylation and activation of phospholipase C-γ. The latter induces an increase in intracellular free calcium that triggers calmodulin/calcineurin/NFAT signaling, thereby promoting cardiomyocyte hypertrophy.

Additionally, some recent studies also have shown that FGF23 hormone can act directly on vascular endothelium to impair endothelium-dependent vasodilation, particularly when expression of the FGF23 co-receptor alpha-klotho is low [30,31,32]. This effect has been traced to diminished vascular production of nitric oxide associated with elevated oxidant production by NADPH oxidase complexes [30]. Consideration should be given to the possibility that the recently reported increase in cardiovascular risk associated with low-normal plasma Pi may be mediated not by low Pi per se, but rather that low Pi may be functioning as a marker for an up-regulation of FGF23 secretion which is the true mediator of risk; this possibility could be evaluated in prospective epidemiology that measures plasma levels of Pi and FGF23. The apparent adverse impact of elevated FGF23 on non-cardiovascular risk conceivably could reflect, at least in part, the ability of FGF23 to act as a growth factor for certain cancers, including possibly prostate cancer [33,34,35,36,37].

## 3. Elevated Phosphate Provokes Mitochondrial Oxidative Stress and Vascular Calcification

Whereas increased production of FGF23 may account for a portion of the excess risk associated with high serum levels and dietary intakes of Pi, elevated Pi may also compromise health—and promote vascular calcification—by boosting oxidant production. Uremia is associated with increased systemic oxidative stress, and there is evidence that elevated Pi levels may act systemically to boost production of superoxide [38]. Indeed, in one recent study which employed serum levels of 8-hydoxy 2-deoxyguanosine as a marker of oxidative stress, this marker rose about *six-fold* when a dietary intake of Pi considered normal for rats (0.6% of diet, associated with a plasma phosphorus level of 4.23 mg/L, within the normal human range) was doubled (leading to plasma phosphorus of 7.14 mg/dL) [39]. This phenomenon may reflect increased superoxide generation by mitochondria, as well as, in at least some tissues including vascular endothelium, increased activation of NADPH oxidase complexes. Studies with isolated mitochondria show that mitochondrial superoxide production rises progressively as medium levels of Pi are increased from normal to supraphysiological intracellular levels; this phenomenon is dependent on intra-mitochondrial transport of Pi, and reflects an induced increase in the transmembrane pH gradient and hence the electric membrane potential delta-psi [40,41,42,43]. An increase in this membrane potential opposes the flux of electrons down the mitochondrial respiratory chain in coupled mitochondria, leading to increased generation of superoxide, particularly at complex III. Co-administration of agents that block mitochondrial Pi uptake or uncouple the respiratory chain inhibit the up-regulatory impact of elevated medium Pi on mitochondrial superoxide generation.

Pi-induced mitochondrial superoxide generation requires intracellular uptake of extracellular Pi via electrogenic transporters of the SLC20 family [23]. This transport entails co-transport of 2 mols Na^+^ for each mol of monovalent Pi, and thus entails a net positive current flux that depolarizes the smooth muscle plasma membrane, provoking Ca^2+^ influx via L-type voltage-sensitive calcium channels; curiously, blockade of these calcium channels prevents Pi from hyperpolarizing mitochondria [23,44]. This calcium requirement might reflect the fact that cytosolic Pi can enter mitochondria via a Pi-MgATP exchanger that requires Ca^2+^ for activity [45].

However, elevations of Pi can also boost superoxide production by NADPH oxidase complexes in vascular endothelium and in vascular smooth muscle [46,47]. How this occurs is uncertain; conceivably, this phenomenon is downstream from a kindling effect of increased mitochondrial oxidant production. The ability of prolonged exposure of osteoblasts to elevated Pi to provoke increased synthesis and secretion of FGF23 reflects an activation of NADPH oxidase that is slow in onset [48]. (Acute elevations of Pi do not provoke an increase in plasma FGF23 levels, but the chronic elevations associated with uremia or high-Pi diets are indeed characterized by elevated plasma FGF23. Hence, the acute adverse effect of a high-Pi meal on endothelium-dependent vasodilation cannot be attributed to FGF23. [17].

As noted, increased vascular calcification is the most characteristic feature of chronic hyperphosphatemia. In vitro, exposure of vascular smooth muscle cells to elevated levels of Pi comparable to those seen in uremia induces a phenotypic change in which these cells take on the properties of osteoblasts; expression of key osteoblast transcription factors—notably cbfa1 (Runx2) and Msx2—is elevated, whereas certain proteins characteristic of vascular smooth muscle, such as smooth muscle actin, are down-regulated [43]. Induction of Runx2 plays an obligatory role in this regard [49]. This phenotypic transition is driven by increased oxidant production, as agents which block mitochondrial oxidant generation prevent it [43]. In one vascular smooth muscle cell line, but not another, inhibition of NADPH oxidase with DPI likewise blocks this phenotypic transition; the varying results might reflect differences in the inherent abilities of the cell lines to express NADPH oxidase [47]. In vivo, administration of the antioxidant agent tempol was found to inhibit vascular calcification in uremic rats [50]. Downstream activation of both ERK1/2 and NF-kappaB appears to play a key role in mediating this effect of oxidative stress [43,47]. The signaling pathway that evokes these activations has not yet been defined. One credible possibility is that oxidant-mediated activation of protein kinase D, expressed in vascular smooth muscle and capable of stimulating the Ras-Raf-MEK-ERK pathway and activating NF-kappaB via IKKbeta, plays a role in this regard [51,52,53,54,55,56].

## 4. Does Magnesium Oppose the Pro-Oxidative Impact of Phosphate on Mitochondria?

Epidemiologically, the association between elevated Pi and increased risk for cardiovascular mortality and progression to end-stage kidney disease in patients with renal dysfunction is blunted or negated in patients whose plasma magnesium (Mg) is in the high–normal range [57,58,59]. Moreover, in vitro, increased medium levels of Mg counteract the ability of elevated Pi to provoke an osteoblastic transition in vascular smooth muscle cells [60,61,62,63,64,65,66]. This phenomenon is dependent on intracellular uptake of Mg. In rodent models of uremic vascular calcification, increased dietary Mg is protective in this regard [67]. Epidemiologically, low serum Mg has been correlated with increased risk for vascular calcification [68,69,70]. These findings have given rise to the concept that good Mg status may act as an antidote to Pi-evoked vascular calcification. A clinical trial assessing the impact of supplemental Mg on progression of vascular calcification in patients with chronic renal diseases is underway [71]. We propose that Mg’s ability to counteract the adverse effect of elevated Pi on vascular smooth muscle reflects an ability of increased intracellular Mg to oppose the impact of elevated intracellular Pi on mitochondrial membrane potential and superoxide generation. To the best of our knowledge, this possibility has not yet been evaluated in the published literature; it could be readily tested in vitro, in intact cells, and in isolated mitochondria. This hypothesis seems credible in light of the fact that normal variations in intracellular Mg are not known to have a major impact on the activity of likely downstream mediators of osteoblastic transition, such as Runx2, ERK1/2, NF-kappaB, and protein kinase D. Moreover, severe Mg deficiency in rodents is known to be associated with oxidative stress [72,73]. It is not clear how Mg might counteract the impact of increased mitochondrial Pi uptake on mitochondrial membrane potential. One testable possibility is that elevated Mg blunts this uptake. As noted, the inner mitochondrial membrane contains a Ca^2+^-dependent transporter which exchanges Pi for a Mg-ATP complex; the immediate impact of an increase in cytoplasmic Mg would presumably be increased mitochondrial export of Pi coupled with increased import of Mg-ATP, counteracting the tendency of hyperphosphatemia to provoke excessive mitochondrial Pi uptake [74]. Clearly, the impact of increased cytosolic Mg on Pi-induced mitochondrial oxidant production warrants study.

Regardless of the mechanism whereby Mg combats the adverse health impact of Pi excess, consuming a primarily plant-based diet, avoiding Pi food additives (as in dietary colas), supplementing with balanced intakes of calcium (which can precipitate Pi in the GI tract, impeding its absorption) and magnesium, and of course avoiding renal damage, could be expected to minimize risk for Pi overload and its negative health effects [24].

## Data Availability

Reviewed studies were retrieved thru PubMed.

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
