# Peer review of "High Intakes of Bioavailable Phosphate May Promote Systemic Oxidative Stress and Vascular Calcification by Boosting Mitochondrial Membrane Potential—Is Good Magnesium Status an Antidote?"

_cells, 2021, doi:10.3390/cells10071744_

Round 1

Reviewer 1 Report

Diets high in bioavailable phosphate is known to promote calcification and oxidative stress.   This short communication review suggested magnesium intake may prevent phosphate mediated calcification. This short communication detailing basic molecular mechanism involving  fibroblast growth factor 23 (FG23) and oxidative stress. Overall the paper is very detailed, providing interesting information to both a lay audience as well as an expert in the field. The clarity is generally outstanding, 

Author Response

Diets high in bioavailable phosphate is known to promote calcification and oxidative stress.   This short communication review suggested magnesium intake may prevent phosphate mediated calcification. This short communication detailing basic molecular mechanism involving  fibroblast growth factor 23 (FG23) and oxidative stress. Overall the paper is very detailed, providing interesting information to both a lay audience as well as an expert in the field. The clarity is generally outstanding, 

Many thanks for your appreciative review of our manuscript.

Reviewer 2 Report

Authors present an interesting study in which in a very clear way they explain the relation between phosphate, cardiovascular disorders and mitochondrial dysfunction. Minor remarks:

1) Please be careful with abbreviations (line 17: FG23),

2) If possible please use the term chronic kidney disease instead of chronic renal failure (line 14); similar: coronary ischemic heart disease (line 56).

3) What do you think about Vitamin D supplementation monitoring, quite popular recently. You mentioned that good monitoring of Pi level involves avoiding Pi food additives and balanced intake of calcium (lines 215-217).

Author Response

Reviewer #2

Authors present an interesting study in which in a very clear way they explain the relation between phosphate, cardiovascular disorders and mitochondrial dysfunction. Minor remarks:

  • Please be careful with abbreviations (line 17: FG23),

             We have fixed this.

  • If possible please use the term chronic kidney disease instead of chronic renal failure (line 14); similar: coronary ischemic heart disease (line 56).

We have changed the wording as suggested.

  • What do you think about Vitamin D supplementation monitoring, quite popular recently. You mentioned that good monitoring of Pi level involves avoiding Pi food additives and balanced intake of calcium (lines 215-217).

Although very high doses of vitamin D are employed in rodents to induce hypercalcemia and vascular calcifications, this would be seen very rarely in humans.  The lowest dose of vitamin D, taken regularly for months, to be associated with toxicity – i.e. hypercalciuria – is about 40,000 IU, a dose far greater than most people would consider taking.  When plasma levels of 25-hydroxycalciferol go up, compensatory downregulation of its conversion to calcitriol prevents an increase of plasma calcitriol (save in a few pathologies such as sarcoidosis).  In fact, the hypercalcemia that can be induced by truly massive doses of vitamin D reflects the fact that a sufficiently high level of 25-hydroxycholecalciferol has agonist activity for the vitamin D receptor.  So, while monitoring of vitamin D levels is prudent to insure that vitamin D status is adequate, such monitoring is unlikely to detect truly excessive levels.

Thank you for your careful attention to our manuscript.

Reviewer 3 Report

The review article by Mark McCarty et al discusses the possible detrimental role of hyperphosphatemia and FGF23 elevation on cardiac and vascular endothelial function and thus possible contributing roles to the cardiovascular risk associated with chronic renal failure. Moreover, it is suggested that an increase of intracellular may be beneficial in opposing the deleterious impact of elevated Pi. Overall, it is clear and well written and is a useful piece of work. I have only a few comments which I hope the authors might consider in their revised manuscript.

  • In the introduction, I would suggest providing some background on Pi metabolism.
  • I wonder where there are findings suggesting other impacts of elevated Pi on vascular endothelium and in vascular smooth muscle cells, in addition to the effects on mitochondrial function and phenotypic transition discussed in the paper.
  • I would suggest discussing studies that do not support the concepts proposed in the paper. I believe it could enrich the discussion and give a more complete perspective on the subject.

Author Response

The review article by Mark McCarty et al discusses the possible detrimental role of hyperphosphatemia and FGF23 elevation on cardiac and vascular endothelial function and thus possible contributing roles to the cardiovascular risk associated with chronic renal failure. Moreover, it is suggested that an increase of intracellular may be beneficial in opposing the deleterious impact of elevated Pi. Overall, it is clear and well written and is a useful piece of work. I have only a few comments which I hope the authors might consider in their revised manuscript.

  • In the introduction, I would suggest providing some background on Pi metabolism.

In line with your suggestion, we have added a new paragraph to the Introduction focusing on phosphate and magnesium.

  • I wonder where there are findings suggesting other impacts of elevated Pi on vascular endothelium and in vascular smooth muscle cells, in addition to the effects on mitochondrial function and phenotypic transition discussed in the paper.

We have discussed the effects of which we are aware.

  • I would suggest discussing studies that do not support the concepts proposed in the paper. I believe it could enrich the discussion and give a more complete perspective on the subject.

We would add a discussion of such evidence if we were aware of it.  If you could point us to such evidence, we will note it in the manuscript.

Many thanks for your thoughtful review of our manuscript.